# Emotional Expectancies and Hostile Attributions as Predictors of Adolescents' Expressions of Emotion with Parents

**Eric W. Lindsey**

Psychology Program, Penn State University Berks Campus; Reading, PA 19610, USA; ewl10@psu.edu;
Tel.: +1-610-396-6033

**Abstract:** The present study examined associations between adolescents' Emotional Expectancies (EE), Hostile Attributions of Intent (HAI), and emotions expressed during interactions with their mother and father. Data were collected from 96 14- to 16-year-olds (27 African Americans, 38 European Americans, and 31 Latinos; a total of 51 girls and 45 boys) and their parents over a period of 2 years. Questionnaires completed by adolescents were used to assess emotional expectancies and hostile attributions of parents' behavior. In both year one and year two, adolescent emotional expressiveness with parents was observed in a laboratory interaction session. Data revealed that both adolescents' EE and HAI in reaction to ambiguous situations predicted their expression of positive emotion with their mother and father 1 year later. EE of happiness were positively related and EE of anger were negatively related to the expression of positive emotion with their mother and father. HAI were negatively related to the expression of positive emotion. Only HAI were related to a higher expression of anger with their mother and father. The findings indicate that HAI and EE represent distinct cognitive-emotional processes that contribute to individual differences in adolescents' expressions of emotion with parents.

**Keywords:** parent–adolescent relationship; emotional expressiveness; emotional expectancies; hostile attributions





## 1. Introduction

Emotions serve a communicative function and are an important indicator of the quality of parent–child relationships during adolescence [1–3]. Within the family, adolescents' expressions of emotion with parents is theorized to play a crucial role in the regulation of autonomy and connectedness [4–6]. Individual differences in patterns of emotional expressiveness with parents have been linked to variations in adolescents' well-being and social adjustment [7,8] and the quality of adolescents' peer relationships [9,10]. Therefore, understanding the concomitant personal and interpersonal characteristics that account for variations in emotional expressiveness across families is an important endeavor.

### 1.1. Theoretical Models

Social information processing models suggest that the expression of emotion is the result of a complex system of internal and external processes [11,12]. Internal processes include various emotion-related memories and cognitive attributions that form an associated network used to plan and execute behavior, including emotion expression [13]. External processes include situational demands that activate emotion regulation strategies and contextualize the interpretation of emotion information. In this way, both the appreciation of the meaning of experiences and the forms of responses that the individual makes to situations are critical steps in both the production and the regulation of emotion.

Social information theory suggests that emotions serve to help children select, arrange, and decipher relevant information in social situations [14,15]. The influence that emotions have on children's effort in interpreting social cues is viewed as a contributing factor to

errors in processing social information. Specifically, the theory argues that when children encounter ambiguous or complex social cues, they automatically call on extant mental structures, or social knowledge, to make sense of the situation [16]. In this way, memories of previous experiences are used to interpret current and novel social events [17,18] by filling the gaps in children's knowledge [16,18]. Both emotional and social-cognitive information are contained in children's memories of past experiences that are brought forward to interpret current social situations [15]. Consequently, understanding the makeup and connections between children's stored affective and social memories may provide a more thorough understanding of the underpinnings of their socio-emotional behavior. Theorists have used the concept of a feedback loop to describe the construction of linkages between emotions and thought processes [19–21]. For instance, Lewis [20] proposed that the foundation of personality patterns is established by a process of identity formation that relies on positive feedback between emotions and cognitive appraisals of experiences. Likewise, sociological views [21] highlight the importance of emotional processes within the family in children's formation of identity that allows them to participate effectively in the wider society. From these perspectives, the internal experience of emotion channels the person's attention toward relevant information pertaining to their goals in a particular situation. That information is used by the individual to make an appraisal about the progress being made toward accomplishing their goals, which in turn generates more emotion that feeds back into the system, generating repeated iterations. This process results in the formation of a multitude of cognition–emotion mental constructs that combine to establish macro personality structures [20,21] linked to a particular style of processing information and engaging with the world [22].

## 1.2. Emotional Expectancies

Researchers have identified Emotional Expectancies [19,23], also referred to as emotion attributions [24] and emotional schemata [25], as a particular form of emotion knowledge made up of expectations concerning the affective responses likely to be prompted by a future event or situation [26]. The conceptualization of EE is consistent with theoretical arguments concerning the heightened personal relevance of emotional experiences, making the antecedents and consequences of such experiences more likely to be stored in long-term memory [15,21]. Cues from new situations that subsequently trigger these memories produce EE in anticipation of and to assist in planning behavioral responses [19,24]. Based on this formulation, EE represent a cognitive structure that serves to connect previous experience to present behaviors [15,21]. Thus, EE may serve a crucial role in the affective responses made during social interactions.

Feedback processes theorized to operate in children's construction of cognitive-emotional structures may also be used to explain individual differences in patterns of association between EE and adolescents' affective responses during social interactions. As Crick and Dodge [16] articulate, children consider their internal emotional states along with other situational cues as relevant sources of social information to be deciphered and understood. Consequently, in any given social encounter, the mood state and emotion experienced by an adolescent become part of the information that is stored in memory for later reference. When a particular memory is triggered in later social situations, odds are high that the mood or emotions that are congruent with that memory will also be recalled, resulting in the past emotional experience influencing the perceptions of current social interaction and contributing to the present emotion expression.

## 1.3. Hostile Attributions

Attributions, a second form of knowledge structure identified in information processing theories [17,27], are defined as the explanations assigned to social events and underlying motives for other people's behavior that are used to facilitate the identification of personal goals and the enactment of appropriate responses [28]. Attributions of intent are a specific type of attribution that can contain accurate or inaccurate information [29,30]. Specifically,

Hostile Attributions of Intent (HAI), have been studied as an inaccurate form of attribution characterized as a deficit in interpreting social cues that contributes to interpersonal difficulties [31]. Dodge's [32] multi-step social information processing model of interpersonal interactions outlines the process by which a tendency to infer hostile intent as an underlying motive of nonhostile and ambiguous social events (i.e., hostile attribution bias) forms a personality-like characteristic that influences social behavior. Empirical evidence linking individual differences in hostile attribution that are consistent with this perspective found that individual differences in children's hostile attribution style are predictive of more hostile and less positive parent–child relationships [33,34].

### 1.4. Adolescence

There is reason to speculate that the transition to adolescence may represent a period in which linkages between cognitive processes and emotions are particularly pronounced. Researchers have found that the onset of puberty is accompanied by an increase in the number and variety of emotion–cognition connections to self-concept and social life [19,21]. The context of parent–adolescent interaction may lend itself to examining such processes, given evidence that negative emotion related to experiences within the family tends to increase during adolescence [3,9]. Early adolescence has also been identified as a period in which the parent's role in providing emotional support declines as peers take priority as sources of excitement and enjoyment [5]. Relatedly, evidence points out that in many families there is a qualitative change in parent–child relationships during early adolescence marked by an elevation in negative emotions and a decline in parental warmth and acceptance [1,6]. Identifying factors that may account for variation in parent and adolescent emotion expression may shed light on processes that are linked to adolescents' socioemotional adjustment.

Considered from a social information processing framework, individual differences in the emotions adolescents express with parents may be accounted for by adolescents' EE and HAI, as well as the interaction between these processes. For example, adolescents who expect to become angry when interacting with their parents have an increased likelihood of expressing anger when interaction does take place. In addition, adolescents who hold HAI for their parents' behavior have a higher probability of expressing anger and a lower probability of expressing positive emotion when interacting with their parents. In turn, youth who hold both negative EE and HAI are especially likely to express more anger, and less positive emotion, than adolescents who have HAI and positive EE, or nonhostile attributions and negative EE. To date, no study has examined both forms of information processing structures, concurrently making it unclear whether EE and HAI make unique or overlapping contributions to adolescents' expressions of emotion with their parents.

### 1.5. Hypotheses

The purpose of this study was to explore associations (see Figure 1) between early adolescents' EE and HAI of parent behavior in relation to the emotions that adolescents express during interaction with their mother and father. Of particular interest was the examination of individual differences in the occurrence of EE and HAI across adolescents. Based on the review of existing literature, four hypotheses were examined:

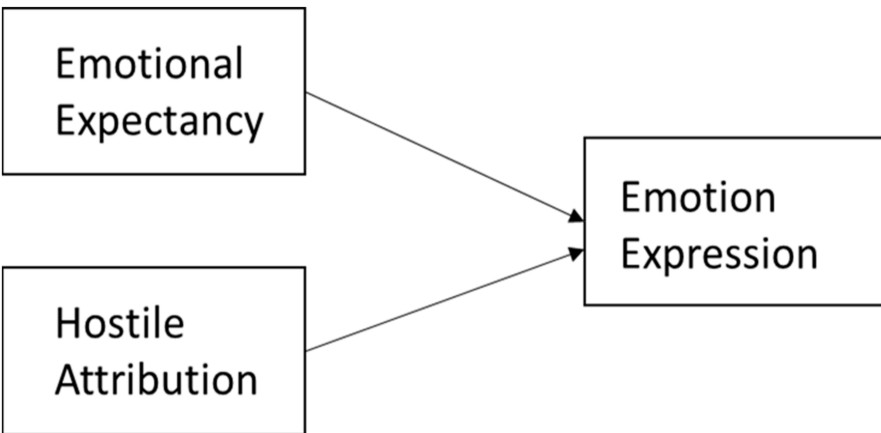

**Figure 1.** Illustration of constructs examined.

**H1.** *Greater EE of happiness would be significantly associated with high levels of expressed positive emotion.*

**H2.** *Greater EE of anger would be significantly associated with high levels of expressed anger.*

**H3.** *Greater HAI would be significantly associated with low levels of positive emotion and (H3) greater HAI would be significantly associated with high levels of anger.*

Although it was of interest to examine interaction effects between adolescents' EE and HAI in predicting expressions of emotion with mother and father, the lack of empirical evidence to guide hypotheses formation means that such analyses were exploratory.

## 2. Materials and Methods

### 2.1. Participants

Data for the present study were collected as part of a larger project examining changes in the quality of parent–child relationships across the transition to adolescence. A sample of 185 families was recruited by telephone during the summer before the adolescent entered seventh grade and followed for two years. Parents of sixth-grade students from 3 public middle schools in a metropolitan northeastern city (population = 197,733) were contacted using rosters obtained from school administrators. Only families in which the adolescent's mother/mother-figure and father/father-figure were living in the home were recruited to reduce confounds in the data. Seventy-two percent of the eligible families agreed to visit the laboratory for the first wave of data collection (adolescent in 6th grade). Families were assessed in 2017 (adolescent in 7th grade) and in 2018 (adolescent in 8th grade), and 52% were retained through the 2017 assessment. The final sample of 96 families had a target child between the age of 13 and 15 ($M$ = 13.87) and an average family income of $M$ = USD 48,250 (range = USD 21,800 to USD 132,200). Parents' length of marriage across families ranged from 11 to 32 years ($M$ = 22 years). Other demographic characteristics of the sample are summarized in Table 1. Analysis comparing the group of families who completed all data collection with those who dropped from the study ($n$ = 89) revealed no significant differences in any demographic characteristics or variables used in the present study $\chi^2(1, 184)$ range = 0.74 to 1.69, ns.

**Table 1.** Demographic characteristics of sample.

| | N (%) | |
|---|---|---|
| Girls | 51 (53%) | |
| Boys | 45 (47%) | |
| Ethnicity | | |
| European American | 35 (37%) | |
| African American | 27 (28%) | |
| Latino | 34 (35%) | |
| Education | Mother | Father |
| Less than high school | 20 (21%) | 14 (15%) |
| Completed high school | 31 (32%) | 37 (38%) |
| Completed some college | 12 (13%) | 10 (10%) |
| Completed college | 27 (28%) | 25 (26%) |
| Graduate-level education | 6 (6%) | 11 (11%) |

*2.2. Procedure*

Each family visited a university laboratory in each year of data collection for approximately 1.5 h, with similar protocols followed at each visit. The visit began with researchers obtaining written consent for participation from the parents and written assent from the focus child. After intake and consent procedures, one family member was taken to a separate room to complete questionnaires on their own while the other two family members were taken to an observation room to participate in an interaction task. After the first interaction task was completed, family members were reorganized so that one family member completed questionnaires while the other two members participated in an interaction task. This procedure was repeated a third time. The three observation sessions consisted of mother–adolescent interaction, father–adolescent interaction, and mother–father interaction. The order in which interaction sessions were conducted was counterbalanced across families. Data from the mother–father interaction task is not considered in this study; therefore, the session is not described in further detail.

During each visit, mother–adolescent and father–adolescent dyads participated in separate 20 min interactions that were videotaped. The interaction sessions were structured around a discussion task developed by Conger and his colleagues [4,35] involving parent and adolescent responding to 14 questions about family life written on cards (e.g., Q1 'What do I do with my parents when we spend time together?'; Q14 'What do my parents usually do when I get into trouble for something?'). Some questions were read by the parent and some by the child (e.g., Q4, read by the child, "What are some rules or things my parents expect me to do or not do? Which of these are fair and which are unfair?"; Q12, read by the mother or father, "If each of us could change anything about our family, what would we like to change? Why? Do we agree or disagree about this?"). Instructions were given to both partners to discuss their responses to the questions on each card, proceeding in consecutive order.

Videotapes of the interaction sessions were subsequently coded using an adapted version of the Emotional Expressiveness coding system [36]. The previous version of the coding system applied scores to every 15 s interaction segment, which failed to capture meaningful instances of discrete emotional expression. Therefore, in the present study, a strategy was adopted to code every 10 s interaction segment. Prior to coding, researchers participated in 40 h of training that included independently rating pre-coded interaction tasks until meeting a criterion of 90% agreement. Different observers rated the mother–child and father–child interaction sessions. For purposes of assessing interobserver reliability, 25% of all videotapes at T1 and T2 were randomly assigned to be rated by a second, independent rater and ratings were compared via Cohen's Kappa [37].

*2.3. Measures*

Family demographics. Family demographic information was obtained from parents. Total income was used as an index of socioeconomic status (SES).

### 2.4. Adolescent Emotional Expectancies (EE) and Hostile Attributions of Intent (HAI)

At T1, adolescents participated in an interview to complete the Child Attribution Measure (CAM [38]) consisting of six hypothetical vignettes. The interviewer read each vignette to the adolescent, describing a situation in which an ambiguous behavior on the part of the parent results in a negative outcome for the adolescent. An example of the content of a vignette is as follows: "Pretend that you and your mom are shopping at the grocery store and that you reach for a candy bar that you want to look at. Your mother tells you that you cannot have it." At the conclusion of each story, the adolescent was asked to respond to three questions: (1) "Why do you think your mother/father would do this?" (open-ended); (2) "How would that make you feel?" (open-ended); and (3) "Was your mother/father being nice or mean?" (closed-ended).

There were two primary versions of the CAM containing six stories each and each primary version had two sub-versions, one in which the stories are worded to be about the mother and one in which stories were worded to be about the father. Within families, adolescents completed a different version of the CAM for their mother and father. Across families, versions of the CAM were alternated so that an equal number of participants completed the same stories that differed only based on which parent was referred to in the vignette, to control for effects of story content. Stories were assigned randomly across families.

A 5-point scale was subsequently used to code adolescents' responses to the first open-ended question (i.e., "Why do you think your mother/father would do this?") with scale anchors reflecting the level of hostility indicated by the adolescent's explanation of their parent's behavior: 5 = hostile, the parent acted purposefully with negative intent (e.g., "She wanted to mess me up."); 4 = moderately hostile (e.g., "He's too busy doing something for himself."); 3 = neutral, the parent acted with harmless intentions or the outcome occurred accidentally (e.g., "She got stuck in traffic."); 2 = moderately prosocial (e.g., "He was cooking dinner for the family."); and 1 = prosocial, the parent intended to behave in a positive way (e.g., "She felt it was dangerous for me.").

Reliability was calculated using intraclass correlations for two coder's ratings of the six responses from 20% of the sample; *r* ranged from 0.83 to 1.00. A separate hostile attribution of intent score was created for both mother and father by summing scores across the six stories. The internal consistency of the composite scales calculated using ordinal alpha was $\alpha = 0.75$ and 0.71 for mother and father, respectively.

A categorical scoring system was used to code adolescents' responses to the second open-ended question "How would that make you feel?", intended to capture emotional self-expectancies. Each response was coded into one of eight categories: (1) happy (e.g., glad, great, good, nice); (2) embarrassed (e.g., self-conscious, humiliated, ashamed, shy); (3) sad (e.g., disappointed, hurt, crushed, down); (4) afraid (e.g., worried, concerned, surprised, anxious); (5) indifferent (e.g., not mad or happy, nothing, wouldn't care, alright); (6) general negative (e.g., bad, discouraged, upset, not too happy); (7) angry (e.g., mad, mean, aggravated, frustrated); and (8) responsive (e.g., "I understand, she had good reason." and "He knew it would benefit me."). Reliability was calculated using Cohen's Kappa for the scores of two coders on 20% of the vignettes; $\kappa = 0.86$ and $\kappa = 0.89$ for mother and father vignettes, respectively. Only the four basic emotion categories were of interest for this study: happy, angry, sad, and afraid. Therefore, four proportion scores were created for adolescents' responses in relation to each parent based on the sum of the responses that were coded in the particular emotion category, divided by the total number of responses. The resulting eight EE scores were: happy, angry, sad, and fear, with mother and with father. Prior to use in analyses, these proportion scores were arcsine-transformed to reduce the skew of the data [39].

### 2.5. Parent-Adolescent Expression of Emotion

Videotapes of the parent–adolescent interaction session at T1 and T2 were coded using a microanalytic coding system to capture adolescents' expressions of emotion. Follow-

ing theoretical formulations, the coding system operationalized emotion expression as occurring across multiple modalities of facial expression, verbal intonation, and nonverbal behavior [21,22]. For each 10 s interval of interaction, coders noted for the adolescent the presence (1) or absence (0) of two categories of emotions. Positive emotion was coded as present if the target displayed positive physical gestures, such as laughing, smiling, hugging, kissing, and/or statements indicating approval, or affection toward the partner (e.g., "good job!"). Anger was coded as present if the target displayed instances of complaints, sarcasm, whining, or nagging, accompanied by negative affect, as well as disruptive behavior such as name-calling, yelling or physical aggression (e.g., pushing, excessive roughness). Evidence for reliability of the coding system has been demonstrated in previous research, and validity has been established by theoretically consistent and significant correlations between the observed emotion scores and adult ratings of children's positive, angry, fearful, and sad temperament characteristics [34,38].

A separate score for each of the two emotion categories, (1) positive emotion and (2) anger, was created for adolescent with mother and adolescent with father, based on the number of intervals in which the particular type of emotion was present out of the total number of intervals in the discussion task session. All proportion scores were subsequently transformed (i.e., arcsine) prior to use in analyses to reduce skew and better approximate a normal distribution [39]. Analyses revealed no significant differences between dyads who did (*n* = 81) and did not (*n* = 15) complete the discussion task in the 20 min allotted on any of the emotion scores *t*(95) range = 0.65 to 1.08, *ns*. Therefore, no adjustments were made to the data based on task completion status prior to use in analyses.

The extensive training in preparation for coding included studying the coding manual, supervised observation of training tapes, and establishing a reliability of κ = 0.85 with the author's coding of training tapes. Upon achieving training criteria, coders were assigned a set of mother-adolescent and father-adolescent tapes to code the emotional expressions of adolescents. No coder was assigned to code a mother–adolescent and father–adolescent dyad from the same family. Periodic "realignment" procedures were implemented after coding every 10 tapes to prevent observer drift, which included a group coding meeting to review the coding manual and have all coders score the same family dyad to compare scores. Final reliability for the coding of adolescent expression of emotion with mother was κ = 0.83 and κ = 0.87 for T1 and T2, respectively. Adolescent expression of emotion with father reliability was κ = 0.87 and κ = 0.89 for T1 and T2, respectively.

### 3. Results

*Preliminary Analyses*

The means and standard deviations of study variables are shown in Table 2. To examine the possibility of differences based on gender and ethnicity, a series of multivariate analyses of variance (MANOVA) were conducted using a 2 (Male vs. Female) $\times$ 3 (European American vs. African American vs. Latino) between-participants design. The results of analysis with HAI as the dependent variable revealed a significant effect for gender (Wilks' $K = 0.78$, $F(2, 177) = 8.05$, $p < 0.01$) but no significant effect for ethnicity (Wilks's $\lambda = 0.98$, $F(3, 390) = 0.57$, *ns*). The follow-up analyses revealed that boys had higher HAI scores than girls ($F(1, 95) = 7.75$, $p = 0.003$, $\eta^2 = 0.13$). The results of the analysis with EE as the dependent variable also revealed a significant effect for gender (Wilks' $K = 0.78$, $F(2, 177) = 11.21$, $p < 0.01$) but no significant effect for ethnicity (Wilks's $\lambda = 0.89$, $F(3, 390) = 0.76$, *ns*). The follow-up analyses revealed that girls had higher happiness and sadness EE than boys, whereas boys had higher EE of anger than girls. The results of analyses with emotion expression as the dependent variable revealed a significant effect for gender (Wilks' $K = 0.85$, $F(2, 177) = 12.34$, $p < 0.001$). The follow-up analyses indicated that girls expressed more positive emotion with their parents than boys ($F(1, 95) = 8.04$, $p = 0.001$, $\eta^2 = 0.15$). There also was a significant effect for ethnicity (Wilks' $K = 0.80$, $F(3, 234) = 7.87$, $p < 0.01$), with follow-up analyses ($F(1, 95) = 5.22$, $p = 0.01$, $\eta^2 = 0.09$) indicating that European American

adolescents express more positive emotion with their parents ($M = 0.37$, $SD = 0.22$) than African American adolescents ($M = 0.30$, $SD = 0.26$).

**Table 2.** Descriptive statistics for adolescent self-reported emotional expectancies, hostile attributions of intent, and observed emotional expressiveness with mother and father.

| | Mother | | Father | |
|---|---|---|---|---|
| | **Boys** | **Girls** | **Boys** | **Girls** |
| | *M* (*SD*) | *M* (*SD*) | *M* (*SD*) | *M* (*SD*) |
| Adolescent EE T1 | | | | |
|   Happy | 0.30 (0.27) | 0.33 (0.33) | 0.27 (0.30) | 0.35 (0.28) |
|   Angry | 0.21 (0.25) | 0.16 (0.23) | 0.20 (0.23) | 0.14 (0.18) |
|   Sad | 0.09 (0.20) | 0.12 (0.16) | 0.10 (0.19) | 0.13 (0.17) |
|   Fear | 0.06 (0.14) | 0.08 (0.15) | 0.05 (0.14) | 0.09 (0.13) |
| Adolescent HAI T1 | 14.46 (2.20) | 12.14 (2.27) | 14.14 (2.12) | 12.08 (2.22) |
| Adolescent emotion expression T1 | | | | |
|   Positive emotion | 0.36 (0.24) | 0.41 (0.24) | 0.30 (0.21) | 0.43 (0.22) |
|   Anger | 0.22 (0.18) | 0.27 (0.20) | 0.24 (0.21) | 0.23 (0.19) |
| Adolescent emotion expression T2 | | | | |
|   Positive emotion | 0.32 (0.19) | 0.37 (0.20) | 0.26 (0.22) | 0.37 (0.21) |
|   Anger | 0.18 (0.20) | 0.22 (0.17) | 0.20 (0.19) | 0.21 (0.20) |

Note: T1 = Year 1; T2 = Year 2; HAI = Hostile Attribution of Intent; EE = Emotional Expectancy.

Bivariate correlations (see Table 3) revealed that adolescents' EE of happiness was significantly negatively associated with HAI toward the corresponding parent. EE of sadness with the father were significantly positively associated with HAI toward the father. EE of happiness with a particular parent was significantly positively associated with the expression of positive emotion with that same parent at both T1 and T2. Similarly, EE of anger with a particular parent was significantly positively associated with the expression of anger with that same parent at both T1 and T2. HAI were significantly positively associated with the expression of anger toward the corresponding parent at both T1 and T2.

The main hypothesis concerning the associations between adolescents' HAI and EE were examined using hierarchical multiple regression analyses. In total, four separate regression analyses were conducted predicting each of the two adolescent emotional expressiveness (i.e., positive emotion and anger) variables with mother and father at T2. Sex and ethnicity were entered in the first step to control for their effects. Adolescent expression of the emotion being predicted during the T1 visit was entered as Step 2. The four adolescent EE scores were entered together in the third step. The adolescent-reported HAI were entered in the fourth step of each analysis.

The product of the respective HAI and EE variables was entered in the fifth step to represent their interaction [40,41].

Significant interactions were probed using standard procedures of calculating simple intercepts and simple slopes [40,42]. The resulting intercepts and slopes produced by these analyses represented the relations between the predictor (adolescent HAI) and outcome (child emotional expressiveness) at lower (1 SD) and higher (+1 SD) levels of the moderator (EE). For interpretational purposes, a significant interaction term reveals that the association between adolescent HAI and the emotional expressiveness outcome variable differs at a higher level of EE than at a lower level of EE. The significance level of the slope indicates the magnitude to which the slope differs from zero at a specific level of EE.

**Table 3.** Adolescent emotional expectancies and hostile attributions as predictors of adolescents' expressions of emotion with mother.

|  | 1 | 2 | 3 | 4 | 5 | 6 | 7 | 8 | 9 | 10 |
|---|---|---|---|---|---|---|---|---|---|---|
| EE with Mother T1 |  |  |  |  |  |  |  |  |  |  |
| 1. Happy |  | 0.16 | 0.13 | −0.11 | 0.37 ** | −0.17 | −0.21 * | −0.07 | −0.21 * | −0.30 ** |
| 2. Angry |  |  | 0.19 * | −0.12 | −0.13 | 0.26 ** | −0.10 | 0.04 | 0.15 | −0.06 |
| 3. Sad |  |  |  | 0.23 * | −0.24 * | 0.17 | 0.20 * | 0.14 | −0.10 | 0.20 * |
| 4. Fear |  |  |  |  | −0.13 | 0.01 | 0.02 | 0.24 * | 0.07 | 0.12 |
| EE with Father T1 |  |  |  |  |  |  |  |  |  |  |
| 5. Happy |  |  |  |  |  | −0.24 * | −0.18 | −0.12 | −0.13 | −0.21 * |
| 6. Anger |  |  |  |  |  |  | 0.20 * | 0.10 | 0.12 | 0.16 |
| 7. Sad |  |  |  |  |  |  |  | 0.14 | −0.10 | 0.20 * |
| 8. Fear |  |  |  |  |  |  |  |  | 0.02 | 0.12 |
| Adolescent HAI T1 |  |  |  |  |  |  |  |  |  |  |
| 9. Mother |  |  |  |  |  |  |  |  |  | 0.22 * |
| 10. Father |  |  |  |  |  |  |  |  |  |  |

|  | 11 | 12 | 13 | 14 | 15 | 16 | 17 | 18 |
|---|---|---|---|---|---|---|---|---|
| EE with Mother T1 |  |  |  |  |  |  |  |  |
| 1. Happy | 0.22 * | 0.11 | 0.12 | −0.11 | 0.30 ** | −0.26 ** | 0.11 | −0.12 |
| 2. Angry | −0.18 * | 0.23 * | −0.15 | 0.03 | −0.16 | 0.28 ** | −0.13 | 0.15 |
| 3. Sad | 0.05 * | 0.13 | 0.06 | −0.10 | 0.07 | −0.15 | 0.03 | 0.06 |
| 4. Fear | 0.07 | 0.10 | −0.10 | 0.08 | 0.12 | 0.07 | 0.14 | 0.10 |
| EE with Father T1 |  |  |  |  |  |  |  |  |
| 5. Happy | 0.18 | −0.15 | 0.22 * | −0.20 * | 0.11 | −0.07 | 0.28 ** | −0.20 * |
| 6. Anger | −0.20 * | 0.08 | −0.26 ** | 0.32 ** | 0.16 | −0.20 * | −0.20 * | 0.43 *** |
| 7. Sad | −0.05 | 0.23 * | 0.06 | −0.09 | 0.05 | −0.04 | 0.09 | 0.06 |
| 8. Fear | 0.07 | 0.06 | −0.10 | −0.12 | 0.05 | 0.10 | 0.01 | 0.10 |
| Adolescent HAI T1 |  |  |  |  |  |  |  |  |
| 9. Mother | −0.05 | 0.31 ** | 0.12 | 0.11 | −0.27 ** | 0.35 ** | 0.11 | 0.12 |
| 10. Father | −0.18 * | −0.11 | −0.19 * | −0.22 * | −0.09 | 0.18 * | −0.29 ** | 0.32 ** |
| Emo. Expression T1 |  |  |  |  |  |  |  |  |
| 11. Positive mother |  | −0.35 ** | 0.42 *** | −0.12 | 0.40 *** | −0.24 * | 0.18 * | −0.13 |
| 12. Anger mother |  |  | −0.15 | 0.30 ** | −0.36 ** | 0.28 ** | −0.13 | 0.26 ** |
| 13. Positive father |  |  |  | −0.20 * | 0.22 * | −0.17 | 0.38 ** | −0.30 ** |
| 14. Anger father |  |  |  |  | −0.16 | 0.10 | −0.22 * | 0.31 ** |
| Emo. Expression T2 |  |  |  |  |  |  |  |  |
| 15. Positive mother |  |  |  |  |  | −0.35 ** | 0.24 * | −0.15 |
| 16. Anger mother |  |  |  |  |  |  | −0.13 | 0.18 |
| 17. Positive father |  |  |  |  |  |  |  | −0.30 ** |
| 18. Anger father |  |  |  |  |  |  |  |  |

Note: T1 = Year 1; T2 = Year 2; HAI = Hostile Attribution of Intent; EE = Emotional Expectancy. * $p < 0.05$, ** $p < 0.01$, *** $p < 0.001$.

As shown in Table 4, in Step 1 of the first regression, child gender and ethnicity accounted for a significant 11% of the variance in adolescents' expression of positive emotion with the mother, with child gender being the primary contributing factor. The beta weights reveal that girls expressed more positive emotion with their mothers than boys. Step 2 of the regression revealed that T1 expression of positive emotion predicted a significant 15% of the variance of T2 positive emotion with the mother. Step 3 revealed that adolescent EE predicted a significant 10% of the variance in positive emotions with the mother, with EE of happiness and anger each independently associated with adolescents' expression of positive emotion with their mothers. EE of happiness were positively associated with expressions of positive emotion and EE of anger were negatively associated with expressions of positive emotion. Step 4 of the regression revealed that higher levels of adolescent HAI contributed a significant 08% of the variance in the expression of positive emotion with the mother, with higher HAI significantly related to lower positive emotion with the mother. The fourth step revealed that these main effects were qualified by a significant two-way interaction between HAI and EE of happiness, and HAI and EE of anger.

**Table 4.** Adolescent emotional expectancies and hostile attributions as predictors of adolescents' expressions of emotion with mother.

| | $\Delta F$ | $\Delta R^2$ | B | SE B | B |
|---|---|---|---|---|---|
| | | | **Positive Emotion (T2)** | | |
| Step 1: | 9.34 ** | 0.11 ** | | | |
|   Child Sex | | | −16.15 | 7.05 | −0.34 * |
|   Child Ethnicity | | | −8.21 | 11.05 | −0.17 |
| Step 2: Expression T1 | 13.41 ** | 0.15 ** | | | |
|   Emotion | | | 8.04 | 4.83 | 0.47 *** |
| Step 3: E. Expect. T1 | 7.41 ** | 0.10 ** | | | |
|   Happy | | | 1.04 | 0.66 | 0.37 ** |
|   Anger | | | 0.95 | 0.72 | −0.22 * |
|   Sad | | | 0.76 | 0.83 | −0.11 |
|   Fear | | | 0.55 | 0.71 | −0.07 |
| Step 4: HAI T1 | 3.56 * | 0.08 * | | | −0.37 * |
| Step 5: Interactions | 5.54 * | 0.07 * | | | |
|   EE Happy × HAI | | | −0.44 | 0.18 | −0.26 * |
|   EE Anger × HAI | | | 0.34 | 0.12 | 0.20 * |
|   EE Sad × HAI | | | −0.24 | 0.63 | −0.07 |
|   EE Fear x HAI | | | 0.14 | 0.37 | 0.11 |
| | | | **Anger (T2)** | | |
| | $\Delta F$ | $\Delta R^2$ | B | SE B | B |
| Step 1: | 2.12 | 0.03 | | | |
|   Child Sex | | | −2.37 | 10.51 | −0.13 |
|   Child Ethnicity | | | −1.15 | 0.85 | −0.15 |
| Step 2: Expression T1 | 15.52 ** | 0.19 ** | | | |
|   Emotion | | | 7.15 | 3.24 | 0.41 *** |
| Step 3: E. Expect. T1 | 2.70 | 0.03 | | | |
|   Happy | | | −0.48 | 0.71 | −0.13 |
|   Anger | | | 0.33 | 0.52 | 0.10 |
|   Sad | | | 0.66 | 0.84 | 0.12 |
|   Fear | | | 0.50 | 0.91 | 0.15 |
| Step 4: HAI T1 | 4.02 * | 0.09 * | | | 0.46 * |
| Step 5: Interactions | 2.08 | 0.02 | | | |
|   EE Happy x HAI | | | 0.74 | 0.88 | 0.13 |
|   EE Anger x HAI | | | 0.69 | 0.83 | 0.06 |
|   EE Sad x HAI | | | 0.47 | 0.05 | 0.03 |
|   EE Fear x HAI | | | 0.54 | 0.13 | 0.09 |

Note: T1 = Year 1; T2 = Year 2; HAI = Hostile Attribution of Intent; EE = Emotional Expectancy. * $p < 0.05$, ** $p < 0.01$, *** $p < 0.001$.

Follow-up analyses revealed that the negative association between HAI and adolescent expression of positive emotion was stronger among adolescents with lower EE of happiness (B = 0.79, SE = 0.12, $p < 0.01$) compared to adolescents with higher EE of happiness (B = 0.34, SE = 0.11, $p < 0.01$). Among adolescents with higher EE of anger, there was a significant negative association between HAI and the expression of positive emotion (B = 0.39, SE = 0.11, $p < 0.001$), whereas, for adolescents with lower EE of anger, there was no significant association between HAI and the expression of positive emotion (B = 0.45, SE = 0.14, $p < 0.01$).

In the second regression (see Table 4) predicting adolescents' expression of anger with the mother, child gender and ethnicity did not contribute significantly to the prediction of anger with the mother. Step 2 of the regression revealed that adolescent expression of anger at T1 significantly predicted 19% of the variance in the expression of anger at T2. Step 3 revealed that adolescent EE did not significantly contribute to the prediction of variance in the expression of anger. Step 4 of the regression revealed that higher levels of adolescent HAI predicted higher levels of adolescent anger with the mother, contributing a significant 09% of the variance to anger with the mother. The fifth step revealed no

significant interactions between HAI and EE of happiness, and HAI and EE of anger in predicting the expression of anger with the mother.

As shown in Table 5, in Step 1 of the third regression, child gender and ethnicity contributed a significant 10% of the variance in predicting the expression of positive emotion with the father. Only child gender was significantly associated with the expression of positive emotion, with the beta weight indicating that girls expressed more positive emotion with the father. Step 2 of the regression revealed that T1 expression of positive emotion predicted the greatest proportion of variance (12%) in T2 positive emotion with the father. Step 3 revealed that adolescent EE predicted a significant 09% of the variance in the expression of positive emotion, with EE of happiness and anger each independently associated with adolescents' expression of positive emotion with their fathers. EE of happiness was positively associated with expressions of positive emotion and EE of anger were negatively associated with expressions of positive emotion. Step 4 of the regression revealed that higher levels of adolescent HAI predicted a significant 05% of the variance in positive emotion with the father, with higher HAI associated with lower positive emotion. The fifth step revealed that these main effects were qualified by a significant two-way interaction between HAI and EE of happiness, and HAI and EE of anger.

**Table 5.** Adolescent emotional expectancies and hostile attributions as predictors of adolescents' expressions of emotion with father.

| | | | Positive Emotion (T2) | | |
|---|---|---|---|---|---|
| | $\Delta F$ | $\Delta R^2$ | B | SE B | B |
| Step 1: | 8.22 ** | 0.10 ** | | | |
| Child Sex | | | −14.58 | 6.95 | −0.35 * |
| Child Ethnicity | | | −7.63 | 10.23 | −0.15 |
| Step 2: Expression T1 | 10.76 * | 0.12 ** | | | |
| Emotion | | | 9.64 | 4.58 | 0.51 *** |
| Step 3: E. Expect. T1 | 7.85 ** | 0.09 ** | | | |
| Happy | | | 0.85 | 0.53 | 0.33 ** |
| Anger | | | 0.91 | 0.44 | −0.28 ** |
| Sad | | | 0.66 | 0.71 | −0.18 |
| Fear | | | 0.55 | 0.67 | −0.17 |
| Step 4: HAI T1 | 3.72 * | 0.05 * | | | −0.39 * |
| Step 5: Interactions | 5.23 * | 0.06 * | | | |
| EE Happy x HAI | | | −0.58 | 0.25 | −0.29 ** |
| EE Anger x HAI | | | 0.30 | 0.10 | 0.21 * |
| EE Sad x HAI | | | −0.44 | 0.80 | −0.04 |
| EE Fear x HAI | | | 0.20 | 0.51 | 0.03 |
| | | | Anger (T2) | | |
| | $\Delta F$ | $\Delta R^2$ | B | SE B | B |
| Step 1: | 2.43 | 0.03 | | | |
| Child Sex | | | −1.77 | 7.11 | −0.15 |
| Child Ethnicity | | | −1.23 | 0.87 | −0.16 |
| Step 2: Expression T1 | 12.70 ** | 0.15 ** | | | |
| Emotion | | | 7.56 | 3.87 | 0.42 *** |
| Step 3: E. Expect. T1 | 2.55 | 0.03 | | | |
| Happy | | | −0.30 | 0.63 | −0.15 |
| Anger | | | 0.38 | 0.41 | 0.18 |
| Sad | | | 0.52 | 0.78 | 0.10 |
| Fear | | | 0.60 | 0.85 | 0.12 |
| Step 4: HAI T1 | 5.15 * | 0.11 * | | | 0.48 * |
| Step 5: Interactions | 1.74 | 0.01 | | | |
| EE Happy x HAI | | | 0.54 | 0.91 | 0.10 |
| EE Anger x HAI | | | 0.29 | 0.73 | 0.01 |
| EE Sad x HAI | | | 0.68 | 0.97 | 0.07 |
| EE Fear x HAI | | | 0.17 | 0.72 | 0.03 |

Note: T1 = Year 1; T2 = Year 2; HAI = Hostile Attribution of Intent; EE = Emotional Expectancy. * $p < 0.05$, ** $p < 0.01$, *** $p < 0.001$.

Follow-up analyses revealed that the negative association between HAI and the adolescent expression of positive emotion was strong among adolescents with lower EE of happiness (B = 0.79, SE = 0.12, *p* < 0.01) compared to adolescents with higher EE of happiness (B = 0.34, SE = 0.11, *p* < 0.01). Among adolescents with higher EE of anger, there was a significant negative association between HAI and the expression of positive emotion (B = 0.39, SE = 0.11, *p* < 0.001), whereas, for adolescents with lower EE of anger, there was no significant association between HAI and the expression of positive emotion (B = 0.45, SE = 0.14, *p* < 0.01).

In the fourth regression predicting adolescents' expression of anger with the father (see Table 5), child gender and ethnicity made no significant contribution. Step 2 of the regression revealed that the adolescent expression of anger at T1 significantly predicted 15% of the variance in the expression of anger at T2. Step 3 revealed that adolescent EE did not predict a significant portion of the variance in the expression of anger. Step 4 of the regression revealed that higher levels of adolescent HAI predicted a significant 11% of the variance in the expression of anger with the father, with higher levels of HAI linked to higher anger with the father. The fifth step revealed no significant interactions between HAI and EE of happiness, and HAI and EE of anger in predicting the expression of anger with the father.

## 4. Discussion

The results of this research provide evidence that adolescents' EE and HAI in response to their parents' hypothetical behavior make independent, as well as interactive, contributions to the emotions that youth express with their parents. Existing theory and research regarding the role of attributions in parent–child relationship quality has focused predominately on attributions of intent [29,32] and has only recently begun to consider how emotional attributions may relate to patterns of behavior in the relationship [36]. Moreover, while the majority of research on links between attributions and behavior has focused on early and middle childhood [16,18], the present study demonstrates that early adolescence represents a developmental period of particular interest for the operation of these processes.

### 4.1. Hypothesized Associations between EE and Emotion Expression

Partial support was found for the hypotheses concerning the relationship between adolescents' EE and emotional expressiveness with parents. Specifically, EE were found to predict only the expression of positive emotion and not the expression of anger with both the mother and father. To the best of my knowledge, this represents the first documentation of a link between early adolescents' EE and emotional expressiveness with their parents. The data revealed a high level of consistency between adolescents' self-reported expectation of emotional experience in hypothetical situations with their parents and the positive emotion that they were observed to express when interacting with their parents. Specifically, as hypothesized, adolescents' EE of happiness and anger that were reported in response to hypothetical stories about their parents was associated with the observed expression of positive emotion during a parent–child interaction session. EE of happiness were associated with a greater expression of positive emotion, whereas EE of anger were associated with a lower expression of positive emotion. These associations are consistent with the tenants of emotional adaptation theory [26] concerning the role that expectations regarding the emotional consequences of some future situation plays in decisions and the enactment of current behavior. That is, the expectations of emotional reactions may serve as a self-fulfilling prophecy concerning what emotions are expressed with parents. Alternatively, the adolescent's expectations of emotions may be built upon a history of experiences with their parent that reflects an ongoing pattern of behavior.

In practical application, the results suggest that EE may be useful for anticipating the positive emotional responses youth will display during interaction with their parents, but that EE may be less accurate in predicting the experience of anger. The link between EE and emotional expressiveness contributes to a greater understanding of how cognitive and

emotional processes may operate in conjunction to impact relationship quality. Instead of forming separate sub-systems of personality dynamics, cognitions and emotions may coalesce to shape complex individual behavior patterns as well as emergent qualities of interaction between partners [15]. The connections between discrete EE and nonparallel emotions suggest that adolescents' mental anticipation of the experience of emotions may serve a preparatory function for social engagement. Specifically, adolescents who have a mental schema of themselves as experiencing positive emotion when interacting with their parents may prime themselves to express positive emotion in the presence of their parents. In contrast, adolescents whose mental schema comprises an expectation to experience anger when interacting with their parents may have difficulty expressing positive emotion.

### 4.2. Hypothesized Associations between HAI and Emotion Expression

Support was also found for hypotheses concerning associations between teens' HAI and emotional expressiveness with parents. Specifically, adolescents who attributed more HAI to a parent's benign behavior in hypothetical stories expressed less positive emotion and more anger when interacting with their parent than adolescents who did not attribute HAI to their parent's behavior. These associations expand upon past empirical results by establishing connections between HAI and adolescents' expression of specific emotions with their parent. Specifically, in keeping with the proposition of attribution theory [18,30], in the present study cognitive biases of perceiving hostility as a motivating force for parental behavior, in the absence of evidence to support such a conclusion, were associated with less positive and more angry emotion expression by adolescents. Thus, HAI appear to be one relevant factor in accounting for individual differences across adolescents in the expression of emotion with parents. That is, any given adolescent's inclination to make HAI of their parent's behavior appears to be an indicator of what level of positive emotion and anger they will express with their parent. Because the exact content of parent and adolescent discussions was not assessed in the current study, it is impossible to know if the adolescent's display of emotion was tied to the topic being discussed in a way that may have been confounded with their HAI. Nevertheless, the findings point to adolescents' HAI as having an influence on adolescents' expression of positive emotion and anger during parent–child interaction.

### 4.3. EE and HAI Associations with Emotion Expression

A major contribution of this study to existing theoretical and empirical evidence is the finding that teens' EE and HAI made independent contributions to their emotional expressiveness with parents. The independent links between HAI and EE and emotional expressiveness support the validity of the conceptual distinction between these two forms of attributions [18,28] and the relevance of this distinction for predicting adolescents' emotional expressiveness. Within the framework of attribution theory, adolescents' HAI and EE represent separate, but related, knowledge structures that may account for individual differences in adolescent expression of emotion. The findings support this proposition in that teens who entered an interaction with their parent expecting to experience happiness were more likely to express happiness. Likewise, adolescents who held HAI of their parent's behavior were less likely to express positive emotion and more likely to express anger when interacting with their parent.

Another noteworthy finding was that adolescents' HAI and EE interacted to predict their emotional expressiveness with parents beyond the individual contributions made by each form of attribution. The pattern of results revealed that adolescents who expected to be happier when interacting with their parent, but who also attributed greater HAI to their parent's behavior, expressed more positive emotion with their parent than adolescents who expected to be less happy with parents and also attributed less HAI to their parent's behavior. In addition, among adolescents with high EE of anger, HAI was significantly associated with a lower expression of positive emotion with parents, whereas among adolescents with low EE of anger, HAI was not related to the expression of positive emotion.

*4.4. Limitations*

It is worth noting that, overall, adolescents expressed low rates of anger with their parents. This finding is consistent with empirical evidence that both teens and parents generally view the parent–child relationship to be positive [5,6]. However, it also raises the concern that the artificial quality of the task in which parent–adolescent interaction was observed may have inhibited participants' expression of anger. Future studies should use more ecologically valid contexts or a range of structured contexts in which to observe parent–adolescent interaction to obtain more accurate assessments of emotions expressed by partners. Furthermore, future research should expand the focus on the adolescent expression of emotion to include a wider range of positive and negative emotions to provide greater insight into cognitive-emotional processes in parent–adolescent relationships.

It is also important to point out that the literature on EE includes studies that focus on children's judgments of how other people are expected to feel [43,44], as well as studies that examine children's expectations regarding their own emotions [45,46]. Both types of expectancies have been linked to the quality of children's social interaction but, in the current study, a decision was made to focus on adolescents' EE regarding their own emotions. This approach was selected to follow theories that an adolescent's cognitions about their own behavior is likely to be more salient to their social interaction than their understanding of the emotional experiences of others. However, it must be acknowledged that how an adolescent thinks about the emotions of another person and what another person is likely to feel in response to behavior that the adolescent displays toward them may play an equally important role in adolescents' choices concerning how to respond to a social partner. Investigations that include both adolescents' self and other EE will help to refine the conceptualization of these constructions and elucidate their role as sources of influence on adolescents' expressed emotions.

There are a number of other limitations to the study reported that should be kept in mind. First, EE were assessed using hypothetical situations, which may have impacted the validity of the content of participants' responses. Future researchers should consider assessing youths' EE in reaction to actual situations experienced with their parent. Second, the examination of possible differences based on ethnicity and sex was limited by the relatively small number of participants in each category. Furthermore, the interpretations of differences between ethnic groups in the present sample should be made with caution, given that potential differences in such areas as family composition, neighborhood quality, and cultural beliefs about emotions were not assessed. The findings of this study need replication with larger samples that allow for appropriate statistical controls. Finally, although predicted longitudinal associations between adolescents' EE and emotional expressiveness with parents were found, as a correlational study these findings do not provide insight concerning causal relations between the variables. To determine the direction of effect among the constructs examined, future experimental studies should be conducted utilizing mood induction techniques with adolescents prior to assessments of EE. Likewise, studies that employ confederate protocols where parents or youth are trained to express patterns of emotions to assess the reactions of the other interactive partner would be beneficial for determining causal links between EE and emotional displays.

*4.5. Conclusions*

In conclusion, the results of the present study provide new evidence on the ways that adolescents' EE and HAI are related to their patterns of emotional expressiveness with parents. Although caution should be exercised when interpreting the findings given the numerous limitations of the study, the results do have several implications for treatment, research, and family policy. First, in relation to treatment, the pattern of associations that were observed implies that efforts to improve emotional communication within parent–adolescent relationships may be misguided if the focus is solely on emotions. Rather, focusing on cognitions held by adolescents, in the form of EE and HAI that have been established based on the history of parent–child interactions, may be crucial to alter current patterns

of emotion expression. Second, in relation to future research, the results suggest that it would be valuable to expand the focus on emotion in the cognitive-contextual framework and explore affective processes involved in adolescents' construction of self-knowledge. Attention should be directed to exploring questions relating to family environments and children's EE, as well as other forms of attributions. Furthermore, the results of the study suggest that answering such questions convincingly may require direct observation of emotion expression and emotion socialization practices in the family, as well as assessing youth cognitive appraisals of family interactions. Finally, in relation to family policy, the findings suggest that programs geared toward promoting higher-quality parent–adolescent relationships should target emotion communication. Educational efforts should be directed at providing parents and adolescents with information about how underlying beliefs about their relationship and their partner's behavior can influence positive and negative emotion expressions. Programs that facilitate parent and adolescent discussion about how they think about their relationship may help improve the quality of the parent–child relationship.

**Funding:** This research received no external funding.

**Institutional Review Board Statement:** The study was conducted according to the guidelines of the Declaration of Helsinki and approved by the Institutional Review Board of Penn State University (IRB #41670; 9 December 2016).

**Informed Consent Statement:** Informed consent was obtained from all parents who participated in the study. Parents also provided informed consent for adolescents' participation and adolescents provided assent for participation.

**Data Availability Statement:** Data for this study are available upon request from the author.

**Acknowledgments:** The author extends his appreciation to Kaylee Grindrod, Samuel Warrick, Tiffany Miller, and the undergraduate research assistants who worked on this study. Gratitude is also expressed to the parents and adolescents who participated in the study.

**Conflicts of Interest:** The author declares no conflict of interest.

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
