# Peer review of "Emotional Expectancies and Hostile Attributions as Predictors of Adolescents’ Expressions of Emotion with Parents"

_adolescents, doi:10.3390/adolescents3010014_

Round 1
Reviewer 1 Report
This paper investigated the relationship between Emotional Expectancies, Hostile Attributions and emotional expressions with parents among adolescents. The topic looks very interesting though the sample size is small. I have several comments as below:
1. I would suggest the authors to divide the introduction part into different sections, for example, theoretical models, Emotional Expectancies, Hostile Attributions, etc.
2. The hypotheses could be presented in a better way with clear wording and organized separately.
3. A flow chart or a figure including the process of the study would be helpful for the readers.
4. Table 1 could be modified better as the “M (SD)s” were redundant.
5. Table 2 needs to be presented in a more organized way or put in the appendix.
6. Please delete the “Primary Analysis” in line 327 and the “discussion” in line 454.
7. A conclusion section should be added.
Reviewer 2 Report
The paper is very interesting. It returns part of the data from a larger project.
I believe that it is worthy of publication, although some improvements are needed.
First, the authors talk about emotions and their importance in the process of identity formation. I believe that on this aspect they should cite some sociologists who have emphasized the importance of the affective relationship between parents and children in the socialization process, such as Parsons.
The information about the participants should be summarized in some tables to make the paper easier to read (lines 146-170).
In the "Procedures" section the authors claim to have used an adapted version of the Emotional Expressiveness coding system. I believe that they need to be clearer on this point, specifying both what this coding system consists of in its original version, what modifications they have made, and the methodological choices they have made underlying their change.
In addition, nowhere is the territorial area of the research reported. We know the nationality of the parents, but we do not know where they live and where the research was conducted. This is a serious shortcoming. The spatial context not only needs to be clearly spelled out, but the description of its characteristics in terms of family policies, views of family roles and emotions are a key framework.
Finally, I believe that in the "Conclusions" the paper needs to better stress how the findings achieved can guide future scientific research and the preparation of new family and social policies.
--
In addition to this, I point out that in the text there are several typos and some inconsistencies with editorial standards. Therefore, in case of acceptance of the paper for publication, I suggest that the paper undergoes careful linguistic and stylistic proofreading.
Round 2
Reviewer 2 Report
I am glad that the authors have taken up most of my suggestions.
In its revised form, the paper is suitable for publication.
However, I recommend that the editing of the text should be concerned. There are still some typos.
For example, bibliographical references are given in round brackets, whereas the journal requires the use of square brackets.
Also, in line 59 the reference to Parsons appears as (Parsons, 1963) instead of [21].
Finally, in the bibliography, there is no bibliographical reference at #22.
